# The Effects of Body Tempering on Force Production, Flexibility and Muscle Soreness in Collegiate Football Athletes

**DOI:** 10.3390/jfmk7010009

**Published:** 2022-01-11

**Authors:** Christopher B. Taber, Roy J. Colter, Jair J. Davis, Patrick A. Seweje, Dustin P. Wilson, Jonathan Z. Foster, Justin J. Merrigan

**Affiliations:** 1Department of Physical Therapy and Human Movement Science, Sacred Heart University, Fairfield, CT 06825, USA; colterr@mail.sacredheart.edu (R.J.C.); davisj3@mail.sacredheart.edu (J.J.D.); sewejep@mail.sacredheart.edu (P.A.S.); wilsond4@mail.sacredheart.edu (D.P.W.); fosterj4@mail.sacredheart.edu (J.Z.F.); 2Rockefeller Neuroscience Institute, West Virginia University, Morgantown, WV 26505, USA; justin.merrigan@hsc.wvu.edu

**Keywords:** tempering, stretching, myofascial release, vertical jumps, isometric strength

## Abstract

There has been limited research to explore the use of body tempering and when the use of this modality would be most appropriate. This study aimed to determine if a body tempering intervention would be appropriate pre-exercise by examining its effects on perceived soreness, range of motion (ROM), and force production compared to an intervention of traditional stretching. The subjects for this study were ten Division 1 (D1) football linemen from Sacred Heart University (Age: 19.9 ± 1.5 years, body mass: 130.9 ± 12.0 kg, height: 188.4 ± 5.1 cm, training age: 8.0 ± 3.5 years). Subjects participated in three sessions with the first session being baseline testing. The second and third sessions involved the participants being randomized to receive either the body tempering or stretching intervention for the second session and then receiving the other intervention the final week. Soreness using a visual analog scale (VAS), ROM, counter movement jump (CMJ) peak force and jump height, static jump (SJ) peak force and jump height, and isometric mid-thigh pull max force production were assessed. The results of the study concluded that body tempering does not have a negative effect on muscle performance but did practically reduce perceived muscle soreness. Since body tempering is effective at reducing soreness in athletes, it can be recommended for athletes as part of their pre-exercise warmup without negatively effecting isometric or dynamic force production.

## 1. Introduction

Coaches and athletes are consistently exploring new methods to improve performance and reduce injury risk during training and competition. To prepare for physical demands, many methods have been employed prior to engaging in the activity, such as stretching, foam rolling, whole-body vibration, self-myofascial release, and dynamic warmups. However, the effectiveness for eliciting acute improvements in performance and recovery typically varies across modalities and exercise selection. Although some methods positively affect a single performance or recovery outcome, human performance practitioners often seek a modality that can positively affect various outcomes across a performance and recovery spectrum (i.e., joint range of motion (ROM), muscle soreness, strength, power). For example, static stretching may cause acute proprioceptive neuromuscular changes that permit a greater ROM [1] but may reduce athletic performances requiring maximal force production or power output (i.e., vertical jumps) [2,3]. For these reasons, additional methods have been sought to simultaneously improve ROM and reduce pain, without impacting subsequent strength or power performances.

Myofascial release is a popular method that is commonly utilized prior to exercise or competition to improve self-perceived feelings of muscle soreness and improve neuromuscular performances [4,5]. Foam rolling is one form of self-myofascial release where the individual actively applies pressure to the extremity via dense foam or plastic roller to elicit changes in the soft tissue [6,7]. The compressive forces applied to the tissue are suggested to modify connective tissue biomechanics and cortical reactions that increased stretch tolerance, subsequently improving ROM and muscle soreness [8]. Yet, according to a recent systematic review, the limited evidence is conflicting and suggests no positive athletic performance effect following foam rolling [4]. In fact, several studies within the systematic review demonstrated decreased performance following foam rolling, which may be exacerbated with longer exposures to foam rolling prior to athletic performances [9,10]. Thus, whether other modes of myofascial release may be preferred prior to strength or power movements is brought to question. 

An emerging method of myofascial treatment is known as body tempering. This method was created by Donnie Thompson in 2014 and involves applying a weighted steel cylinder to a muscle group to passively manipulate the tissue under the device. This treatment has been suggested to improve tissue tolerance, elasticity, and reduce pain [11,12]. These mechanisms are posited to be similar to those experienced with foam rolling, despite the differences in application between modalities. Foam rolling is an active process requiring the athlete to move across the roller using a self-selected amount of pressure by adjusting the amount of their bodyweight placed on the device. Body tempering is a passive process employed by placing the device of a known mass on the athlete. Thus, one advantage of body tempering compared to foam rolling is the athlete being able to relax under the applied load. Additionally, by knowing the mass of the device being used, coaches and practitioners can control the pressure and ensure consistency within and between athletes. 

Despite the popularity of body tempering, particularly in strength and power athletes, no study has investigated the use of body tempering for simultaneously enhancing soreness, ROM, and strength or power in athletes. Thus, to support the use of body tempering by strength and conditioning practitioners, scientific investigation is warranted. Therefore, the purpose of this study is to examine the acute effects of body tempering on soreness, ROM, vertical jump performance, and maximal strength in D1 football players.

## 2. Materials and Methods

### 2.1. Experimental Approach to the Problem

To investigate the effects of body tempering on performance, a randomized counterbalanced repeated measures design was employed. The subjects reported to the laboratory three times for measurements of baseline data and two experimental sessions (tempering or stretching). These sessions were separated by seven days to prevent fatigue and the effect of order on performance. Participants maintained their normal football training schedule during the collection of this study. 

### 2.2. Subjects 

This study included 10 male (mean ± SD; age, 20.0 ± 1.5; height, 1.88 ± 0.05 m; body mass, 131.0 ± 12.1 kg) Division I National Collegiate Athletics Association (NCAA) offensive and defensive linemen from the Sacred Heart University American football team. All subjects had prior strength training experience (4–13 years) and were familiar with body tempering (at least one body tempering treatment prior to testing). Subjects’ medical histories were free of metabolic, cardiovascular, endocrine, thermoregulatory, and musculoskeletal complexities (i.e., no lower body musculoskeletal injuries in the previous six months that would impede ability to perform current testing). Additionally, no subjects below the age of 18 were admitted to this study. All subjects completed the institutional review board approved informed consent and were provided the chance to ask any questions or request any clarification about the procedures prior to testing. This study was approved by the university’s institutional review board and all procedures were in concordance with the Declaration of Helsinki.

### 2.3. Procedures 

Subjects completed a total of three 30-min testing sessions separated by one week. In the time between sessions, the athletes were allowed to train in accordance with their football training protocol, which was similar across all athletes in the current study. The first session consisted of baseline testing and served as a familiarization session. However, due to the potential for fatigue from performing the isometric mid-thigh pull (IMTP) during later sessions, the first session served as a baseline for the IMTP. In the second session, athletes were randomly assigned to perform either body tempering (*n* = 5) or stretching (*n* = 5) protocols. Then the athletes performed the remaining protocol the following week during the third session. During each experimental session, the athletes performed the same battery of testing prior to and immediately following the protocol (body tempering or stretching) in the following order: perceived soreness levels, ROM (ROM; Thomas Test, straight leg raise test, Eli’s test, and 90–90 extension test), squat jump (SJ), countermovement jump (CMJ), and IMTP (only post-protocol). A warmup protocol was conducted on each testing session, after ROM and prior to the jump testing. The warmup protocol consisted of 20 jumping jacks, 10 squat jumps, and 10 total reverse lunges. Each session was supervised by a certified athletic trainer and/or strength and conditioning specialist.

#### 2.3.1. Body Tempering and Stretching Protocols

The stretching protocol involved two lower body stretches performed in the following order: the standing quad stretch and forward reaching hamstring stretch with their foot on a milk crate. Each stretch was held statically, at moderate discomfort, for 30 s and was repeated 3 times. The total active stretching time was six minutes and athletes were allotted 10 s of passive rest between stretching sets. For body tempering, athletes laid down on the floor and a 59 kg tempering roller was placed on the athlete’s thigh by an athletic trainer. The roller was placed on the proximal, middle, and distal muscle belly of the quadriceps and hamstrings for 30 s in each position, totaling six minutes between both legs. Tempering was completed for one leg at a time. First, athletes were in the supine position and tempering was completed on the quadriceps. Then, athletes assumed the prone position for body tempering on the hamstrings. Athletes were given 10 s of passive rest between body tempering sets. 

#### 2.3.2. Soreness

Soreness was assessed using a visual analog scale. Subjects were given a line on a piece of paper measuring 100 mm long. The far left and right sides of the scale were labeled as “no soreness” and “extreme soreness”, respectively. Subjects were instructed to put an X on the line corresponding to their current perceived soreness. The measured distance from the left end of the line to the center of the X was used to quantify the subjects perceived level of soreness. 

#### 2.3.3. Range of Motion (ROM)

Measures of ROM included standard clinical tests (Thomas Test, straight leg raise test, Eli’s test, and 90–90 extension test). For each test, the ROM was assessed digitally with the use of Kinovea (Kinovea.org, Version 0.9.3). Subjects had their mid axillary, greater trochanter, lateral epicondyle, and lateral malleolus marked with athletic tape and a sharpie upon beginning the study for that day. Using these markers ensured that the same anatomical landmarks were used for assessment of ROM. The first student researcher then helped the subject maintain the appropriate testing position so the third student researcher could take a picture of the subject in the testing position, which was then used to digitally assess the subject’s ROM. Digital photos of the subjects were taken using a Sony a6000 (Sony Electronics, Minato City, Tokyo, Japan). Distance from the treatment table to the camera sensor was 93 inches, distance from the floor to mid sensor height was 48 inches, the distance from the floor to the top of the treatment table was 28 inches, and the camera was perpendicular to the midline of the treatment table. Focal length of the camera lens was set at 20 mm.

#### 2.3.4. Static and Countermovement Jumps

Following ROM testing, subjects completed maximal SJ and CMJ vertical jump testing consisting of two warmup jumps with 50 and 75% of their perceived maximum followed by four maximal jump trials of each style. For the SJ, arm motion was neutralized as the athletes held a ¼ inch Polyvinyl chloride (PVC) pipe across their upper back. Subjects were instructed to squat down to 90° of knee flexion and remain stable during a 3 s countdown. On the command “jump”, subjects were instructed to jump as high as they safely could and land back on the force plates. Trials were disqualified and repeated if the subject began their jump by squatting down more or if they did not land on the force plates properly. For the CMJ, the trial jumps were repeated before the maximal attempts. When given the command “go”, subjects were allowed to squat down to any depth and use any arm motion they deemed necessary to achieve their maximal CMJ. Trials were disqualified and repeated if the subject did not land on the force plates properly. All data were collected on a dual force plate (ForceDecks, Vald Performance, Newstead, QLS, Australia) set up. Jump height and peak power were used for analysis. 

#### 2.3.5. Isometric Mid-Thigh Pull

After vertical jump testing, subjects completed a maximal IMTP in a Kairos Strength IMTP rack (Kairos, Murphy, NC, USA). Subjects were given two warmup trials with 50 and 75% of their perceived maximum followed by two trials at 100% effort during which their hands were strapped to the bar with lifting straps and athletic tape. All subjects completed pulls with a knee flexion angle of 125–135° and hip angle of 145–155°. The bar height from session one was used for all subsequent measurement sessions. All maximal strength data were collected on a dual force plate (ForceDecks, Vald Performance, Newstead, QLS, Australia) collected at 1000 Hz. Maximal force production was collected for data analysis.

### 2.4. Statistical Analysis

Data are displayed as mean ± SD. Data were considered normal according to Shapiro–Wilks tests and histograms. A 2 (Time) × 2 (Group) analysis of variance (ANOVA) was run for each metric (ROM, soreness, squat jump, vertical jump, IMTP) to determine the effects of body tempering and stretching. Since the IMTP was not conducted at pre-testing, due to potential fatigue, the baseline data were used in comparison to post-testing data. Post hoc analyses with Bonferroni corrections were conducted following any significant univariate effect. Statistical significance was set at *p* ≤ 0.05 for all analyses. All analyses were computed using R (Version 4.0). A Cohen’s D effect was obtained to determine the magnitude of the effects between the two groups.

## 3. Results

There was no statistically significant protocol by time interactions or main protocol effects for any metric (Table 1). All group (tempering and stretching) mean and standard deviations at pre- and post-testing, as well as percent changes and effect sizes are displayed in Table 2. Visual display of individual responses can be seen in Figure 1.

## 4. Discussion

This study examined the effects of body tempering versus stretching on ROM, soreness, and lower body force and power production. Although no statistically significant differences were found between conditions, large positive effect sizes were present for straight leg raise ROM (*d* = 0.84) and large negative effects for a reduction in soreness (*d* = −0.94) following body tempering. Static stretching improved 90–90° ROM (*d* = 1.0) with large negative effects on countermovement jump peak power (*d* = −1.22). It appears that neither tempering nor stretching have significantly negative performance effects prior to exercise, but both may improve ROM and tempering may reduce soreness prior to exercise.

The perception of pain or soreness in athletes is a subjective variable that has the potential to inhibit performance [13]. Since soreness may be a limiting factor in the weight room and on the playing field, training staff often look for modalities that can reduce perceived levels of pain in hopes to improve performance. Although not statistically significant, the tempering device reduced perceived soreness by 42.0%, whereas stretching reduced soreness by 10.9%. The exact physiological mechanism for this reduction in soreness is not fully elucidated, but it is likely a combination of mechanical and neurophysiological mechanisms [14]. Body tempering differs from other modalities in that it is a passive modality that involves high amounts of pressure due to the known mass (e.g., 59 kg) of the tempering device. Athletes that report lower levels of soreness may exhibit better performance during training sessions [15]. Thus, due to the positive effects of body tempering on muscle soreness, we anticipated improved ROM and performance measures.

Although other soft tissue modalities, such as stretching, and various massage techniques have been shown to increase ROM, less is known about body tempering effects [14]. The current results on ROM suggest that static stretching caused a small percent change in all ROM variables (2.9–6.0%), while body tempering elicited a slightly greater change in the straight leg raise (6.7 vs. 3.4%) and had a small effect (1.22%) in the Eli test. The effects of static stretching on ROM are to be expected according to prior literature [16,17]. The large effect size in favor of improved straight leg raise ROM from body tempering was likely due to the similarly extended leg position during body tempering. It is possible that targeted static body tempering may elicit ROM changes in specific positions, but it is unknown whether similar effects may occur with the use of dynamic body tempering across various positions. Overall, the evidence for body tempering to increase ROM is conflicting depending upon the ROM assessment, which may be mirrored by effects of body tempering across performance assessments.

Although it was originally believed that alterations in fascial pliability may lead to deficits in muscle coordination and force production [18], some prior research has found no subsequent decrease in force production following self-myofascial release [4,19,20]. In the current investigation, neither tempering nor stretching statistically reduced countermovement and squat jump height or power. However, there were moderately negative effects on countermovement jump height from both stretching (*d* = −0.76) and tempering (*d* = −0.75) and large negative effects on countermovement jump peak power following static stretching (*d* = −1.22) compared to small effects following tempering (*d* = −0.20). The amount of time spent stretching was limited to 30 s per bout and held until mild discomfort, which may have prevented large negative effects that have been observed in other studies [21]. Since no other studies have examined long durations of applying body tempering, the duration of applying body tempering should be slowly increased while monitoring each athlete’s tolerance and response.

Along with reductions in dynamic force and power production, there have been documented reductions in isometric force production following static stretching [21]. Other studies have investigated myofascial release techniques on strength and documented no alterations in performance [6,22,23]. Grabow et al. applied a constant pressure rolling device of varying loads, low (15%), moderate (21%), and high (27%) percent of body mass, on the quadriceps muscle group [22], and found an insignificant effect on muscle strength with improved ROM [19]. Our study is in alignment with earlier studies with insignificantly small to moderate decreases in force production after body tempering (*d* = −0.23) and static stretching (*d* = −0.51), respectively. The highest amount of pressure mentioned in the aforementioned literature was 27% of the individual’s body mass, which was only a 12% variation from the lowest load used in the study. However, our investigation used a heavier body tempering device (59 kg), closer to 50% of the current athlete’s body mass, which is more commonly used by practitioners and may even exceed 50% of body mass for smaller athletes. This provides further evidence as to the lack of effect of body tempering on force production capabilities throughout a wider range of pressure from the body tempering device.

While this investigation provides some practical aspects of using body tempering pre-training, some considerations should be given to implementation. First, our subjects were not naïve to this treatment and had received body tempering in the past. Secondly, these athletes were American football linemen with an average body mass of 131 kg. When working with other sporting athletes, consideration should be given to individual response to loading in relation to the weight of the roller, time instituted, and type of tempering used (static vs. dynamic). Finally, this was an acute study examining pre-training effects of body tempering; caution should be given to extrapolating out chronic effects on ROM, strength, and power production. For example, since body tempering affected perceived muscle soreness, it is possible that the athletes in the current study may have moved more efficiently, according to prior research [15]. By doing so, the small improvements in technique and performance over time may result in improved athletic performances over time.

## 5. Conclusions

The results of this study demonstrated that tempering can be an effective way to reduce soreness in strength and power athletes. Since athletes may report a reduction in soreness, allowing for improved performance during training or competition, body tempering can be used before training or competition to prepare athletes for the task at hand. Body tempering is a relatively quick intervention, only requiring three minutes per limb, which can allow for multiple athletes to receive the intervention in a short period of time. It is recommended that the tempering device be handled by a trained individual who can monitor the pressure being applied by the device and appropriately place the device. The roller used in this study was 59 kg, which corresponded to ~50% of the athlete’s body mass. Athletes from different sports and those of different sex may not tolerate or receive the same benefits from this aggressive ratio. Future studies should be conducted using different athletes and different tempering device weights, as well as tempering other body parts.

## Figures and Tables

**Figure 1 jfmk-07-00009-f001:**
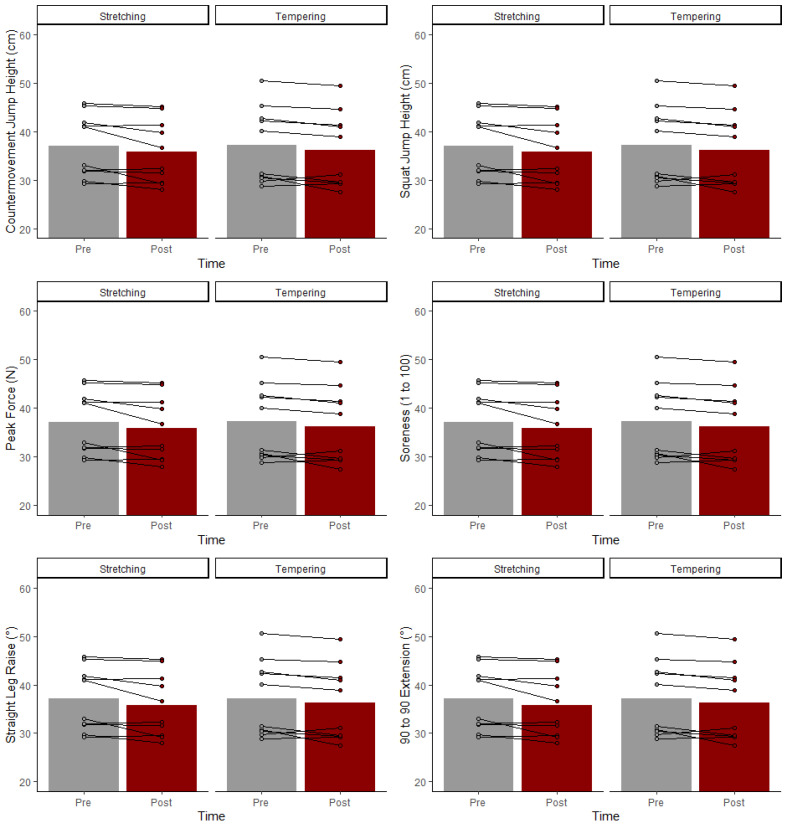
Group and individual responses for select testing metrics for stretching and body tempering.

**Table 1 jfmk-07-00009-t001:** Main time and interaction effects (F statistic, *p*-value).

Variable	Protocol Effect	Protocol × Time Interaction
Straight Leg Raise (°)	(0.003, 0.958)	(0.283, 0.598)
Thomas Test (°)	(0.018, 0.895)	(0.309, 0.582)
90–90 Extension (°)	(0.015, 0.903)	(1.452, 0.236)
Eli Test (°)	(0.064, 0.801)	(0.064, 0.801)
Soreness (mm)	(1.379, 0.248)	(1.230, 0.278)
SJ JH(cm)	(0.010, 0.921)	(0.047, 0.829)
SJ PP (W)	(0.003, 0.953)	(0.003, 0.953)
CMJ JH(cm)	(0.012, 0.913)	(0.004, 0.951)
CMJ PP (W)	(0.010, 0.921)	(0.047, 0.829)
Peak Force (N)	(0.211, 0.649)	(0.211, 0.649)

Note: SJ JH, squat jump height; SJ PP, squat jump peak power; CMJ JH, countermovement jump height; CMJ PP, countermovement jump peak power; peak force, isometric mid-thigh pull peak force.

**Table 2 jfmk-07-00009-t002:** Group averages presented as mean (M) ± standard deviation.

Variable	Group	Pre	Post	%Change	Effect Size
Straight Leg Raise (°)	Tempering Stretching	62.4 ± 3.37 63.3 ± 7.69	66.6 ± 5.89 65.5 ± 6.0	6.73 3.48	0.84, large 0.41, small
Thomas Test (°)	Tempering Stretching	86.5 ± 12.01 83.9 ± 13.36	87.40 ± 10.56 89 ± 11.66	1.04 6.08	0.12, negligible 0.58, moderate
90–90 Extension (°)	Tempering Stretching	150.9 ± 4.93 148.2 ± 7.96	150.6 ± 5.44 152.80 ± 6.94	0.20 4.60	−0.04, negligible 1.0, large
Eli Test (°)	Tempering Stretching	114.7 ± 11.09 112.80 ± 12.0	116.1 ± 9.92 116.1 ± 13.97	1.22 2.93	0.27, small 0.57, moderate
Soreness (mm)	Tempering Stretching	43.0 ± 26.66 43.4 ± 16.89	23.90 ± 14.16 37.9 ± 17.51	42.06 10.99	−0.94, large −0.44, small
SJ JH(cm)	Tempering Stretching	32.29 ± 6.48 31.82 ± 6.02	32.29 ± 6.19 31.84 ± 5.40	0.02 0.54	−0.07, negligible 0.01, negligible
SJ PP (W)	Tempering Stretching	6301.6 ± 729.4 6301.6 ± 728.9	6242.3 ± 579.7 6266.9 ± 577.0	0.64 0.23	−0.19, negligible −0.12, negligible
CMJ JH(cm)	Tempering Stretching	37.22 ± 7.86 37.11 ± 6.52	36.25 ± 7.78 35.86 ± 6.59	2.59 3.37	−0.76, moderate −0.75, moderate
CMJ PP (W)	Tempering Stretching	6471.4 ± 691.5 6494.2 ± 627.1	6423.6 ± 552.91 6361.70 ± 581.67	0.48 1.99	−0.20, small −1.22, large
Peak Force (N)	Tempering Stretching	4202 ± 534 4202 ± 534	4113 ± 373 3960 ± 690	1.34 5.55	−0.23, small −0.51, moderate

Note: SJ JH, squat jump height; SJ PP, squat jump peak power; CMJ JH, countermovement jump height; CMJ PP, countermovement jump peak power; peak force, isometric mid-thigh pull peak force.

## Data Availability

The data presented in this study are available on request from the corresponding author. The data are not publicly available due to it being private athlete data.

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
