# Peer review of "The Effects of Body Tempering on Force Production, Flexibility and Muscle Soreness in Collegiate Football Athletes"

_jfmk, 2022, doi:10.3390/jfmk7010009_

Round 1
Reviewer 1 Report
This is a very well written article and I recommend for publication. The only minor edit I would suggest is removing Figure 1 as all the data is presented in the Tables and Figure 1 is redundant.
Author Response
Reviewer #1
This is a very well written article and I recommend for publication. The only minor edit I would suggest is removing Figure 1 as all the data is presented in the Tables and Figure 1 is redundant.
Thank you for your kind words about our manuscript.
We included table 1 to highlight the individual differences from the athletes. We provided the mean data in the tables and showed within the figures how each athlete changed due to the protocol. We feel this is valuable with this sample size and to highlight individual variation from the same stimulus. For this reason, we would like to include it in the full manuscript.
Reviewer 2 Report
The scientific problem proposed in this work is relevant, the introduction presents the state of the art adequately, the experimental design is appropiate but there exist some methological problems.
1. Poor external validity. The authors intend to assess the effect of the myofascial training on the performance of a soccer player. But they use tests which does not allow an adequate assessment of performance from my point of view. I consider that in a soccer, performance should be assessed through skills such as kicking, sprint etc.
What is the justification of using some skills such as a vertical jump or a maximum isometric strength for testing the performance of a soccer player?
2. The experimental design is adequate but the statistical tests used do not allow finding differences between protocols. The authors have applied an ANOVA test for analyzing the main effect of the "Protocol" but have not compared whether the increases between pretest in posted are statistically different between protocols for each of the dependent variables.
The authors say in line 236: "In the current investigation, neither tempering nor stretching statistically reduced counter-movement and squat jump height or power".
But on page 238 they say: "However, there were moderately negative effects on countermovement jump height from both stretching (d = -0.76) and tempering 239 (d = -0.75) and large negative effects on countermovement jump peak power following static stretching ( d = -1.22) compared to small effects following tempering (d = -0.20) "
This is difficult for the reader to understand because it is said first that there are no differences and after that there exist . If I have understood well the meaning of this clauses in document.
My recommendation is that a comparison of the means of the increments (between the pretest and the posttest) between the two protocol should be made for each metrics. (pair wise comparison using bonferroni correction).
What is the opinion to the authors in relation with this matter?
Author Response
Reviewer #2
The scientific problem proposed in this work is relevant, the introduction presents the state of the art adequately, the experimental design is appropriate but there exist some methodological problems.
Thank you for your review. Below are some clarifications which will help justify the reasons we chose to include various test and statistical methodologies.
- Poor external validity. The authors intend to assess the effect of the myofascial training on the performance of a soccer player. But they use tests which does not allow an adequate assessment of performance from my point of view. I consider that in a soccer, performance should be assessed through skills such as kicking, sprint etc.
In lines 87-90 it was stated that these were American football linemen. These methods are currently in use by football linemen however little research has been completed on their efficacy. We chose these methods to examine the acute effects of body tempering before training. Because little is known about the effects of this device in a controlled setting we chose range of motion, soreness, lower body power, and maximal strength production to cover many aspects of performance. These tests are commonly done in a strength and conditioning setting and have applications prior to training and for long-term development of the athletes investigated.
What is the justification of using some skills such as a vertical jump or a maximum isometric strength for testing the performance of a soccer player?
These tests were chosen because they are commonly found in the literature for examination of lower body power (countermovement jumps) and maximal force production (isometric mid-thigh pull). These are two important factors for this specific population of American football lineman. Additionally, the training used to develop these athletes are maximal strength training and power training including plyometrics. Because of this, we wanted to examine the effects of body tempering on these performance methods to help inform coaches for practical implementation.
The experimental design is adequate but the statistical tests used do not allow finding differences between protocols. The authors have applied an ANOVA test for analyzing the main effect of the "Protocol" but have not compared whether the increases between pretest in posted are statistically different between protocols for each of the dependent variables.
The ANOVA was chosen after consulting a research statistician at our university. There are three possible outcomes that can arise from the use of an ANOVA in the manner conducted. We can have either a group, a time, or a group by time (interaction effect). In this study, none of these were found during the examination of the data. If pretest and posttest were different we would have observed a time effect or if one group was different we would have observed a group effect. This did not occur in our investigation. Thus, we did not do any post hoc pairwise comparisons or t-tests since no main effects were statistically significant.
The authors say in line 236: "In the current investigation, neither tempering nor stretching statistically reduced counter-movement and squat jump height or power".
But on page 238 they say: "However, there were moderately negative effects on countermovement jump height from both stretching (d = -0.76) and tempering 239 (d = -0.75) and large negative effects on countermovement jump peak power following static stretching ( d = -1.22) compared to small effects following tempering (d = -0.20) "
This is difficult for the reader to understand because it is said first that there are no differences and after that there exist . If I have understood well the meaning of this clauses in document.
We did not observe any statistical significance during the analysis of the squat or countermovement jumps which was analyzed with the ANOVA. We ran cohen’s d effect sizes to examine the magnitude of effect. This has been considered a practical effect and is often used when sample sizes are limited. So you may report that a p-value was non-significant with a large magnitude of effect (cohen’s d) indicating important information though the alpha level was not obtained. There were certain individuals that experienced decreases in these performances based on individual responses included in figure 1. However, since these findings were not statistically significant they should be used with caution until further investigated
My recommendation is that a comparison of the means of the increments (between the pretest and the posttest) between the two protocol should be made for each metrics. (pair wise comparison using bonferroni correction).
What is the opinion to the authors in relation with this matter?
We chose to run the ANOVA based on consultation with our research statistician. We could have run multiple comparisons with Bonferroni corrections however due to the amount of t-test we would have to complete we would find the same results due to the change in alpha level with each subsequent test. With each subsequent t-test, you run the risk of committing a type 1 error which must be accounted for with a correction. An ANOVA controls for this allowing for the researchers to be more confident with results if a true result is found. If we only had a few variables multiple t-tests would be a viable option but for our analysis running the ANOVA with cohen’s d effect sizes was the most appropriate choice. Furthermore, as stated previously because the main effect was not present we did not follow up with any additional tests.